# An Optimized Dual Extraction Method for the Simultaneous and Accurate Analysis of Polar Metabolites and Lipids Carried out on Single Biological Samples

**DOI:** 10.3390/metabo10090338

**Published:** 2020-08-19

**Authors:** Joran Villaret-Cazadamont, Nathalie Poupin, Anthony Tournadre, Aurélie Batut, Lara Gales, Daniel Zalko, Nicolas J. Cabaton, Floriant Bellvert, Justine Bertrand-Michel

**Affiliations:** 1Toxalim (Research Centre in Food Toxicology), Université de Toulouse, INRAE, ENVT, INP-Purpan, UPS, 31027 Toulouse, France; joran.villaret-cazadamont@inrae.fr (J.V.-C.); nathalie.poupin@inrae.fr (N.P.); daniel.zalko@inrae.fr (D.Z.); nicolas.cabaton@inrae.fr (N.J.C.); 2MetaboHUB-MetaToul-Lipidomics Core Facility, Inserm U1048, 31432 Toulouse, France; tournadre.anthony@sfr.fr (A.T.); aurelie_batut@orange.fr (A.B.); 3MetaboHUB-MetaToul, National Infrastructure for Metabolomics and Fluxomics, 31077 Toulouse, France; jeanblan@insa-toulouse.fr; 4Toulouse Biotechnology Institute, Université de Toulouse, CNRS, INRAE, INSA, 31400 Toulouse, France

**Keywords:** lipidomics, metabolomics, multi-omics analysis, sample preparation, dual extraction, hepatotoxicity

## Abstract

The functional understanding of metabolic changes requires both a significant investigation into metabolic pathways, as enabled by global metabolomics and lipidomics approaches, and the comprehensive and accurate exploration of specific key pathways. To answer this pivotal challenge, we propose an optimized approach, which combines an efficient sample preparation, aiming to reduce the variability, with a biphasic extraction method, where both the aqueous and organic phases of the same sample are used for mass spectrometry analyses. We demonstrated that this double extraction protocol allows working with one single sample without decreasing the metabolome and lipidome coverage. It enables the targeted analysis of 40 polar metabolites and 82 lipids, together with the absolute quantification of 32 polar metabolites, providing comprehensive coverage and quantitative measurement of the metabolites involved in central carbon energy pathways. With this method, we evidenced modulations of several lipids, amino acids, and energy metabolites in HepaRG cells exposed to fenofibrate, a model hepatic toxicant, and metabolic modulator. This new protocol is particularly relevant for experiments involving limited amounts of biological material and for functional metabolic explorations and is thus of particular interest for studies aiming to decipher the effects and modes of action of metabolic disrupting compounds.

## 1. Introduction

Metabolomics and lipidomics are relatively novel technologies [1], which allow access to a complex and comprehensive snapshot of the status of a biological system (cell, tissue, organism, or body fluid) at a given time [2]. Compared to other “omics” technologies (transcriptomics, proteomics…), metabolomics and lipidomics are considered to provide information that is closer to the actual phenotype and, therefore, more relevant for interpreting phenotypic changes [3]. These technologies are relevant and promising tools for studies related to metabolic disruption in many different contexts, including disease diagnosis [4], toxicology [5,6], nutrients’ effects [7] as well as in the field of plant sciences [8]. However, to explore changes occurring simultaneously at the level of many different metabolic pathways, it is essential to be able to perform unbiased measurement of a maximal number of metabolites on a given biological sample, which is often of limited size/volume [9]. Increasing the coverage of both the metabolome and lipidome is challenging as it requires, most of the time, to combine different sample preparations and analytical techniques [10]. Indeed, the recovery of lipids and polar metabolites involves distinct, costly, and time-consuming extractions [11,12]. In classically used protocols, polar metabolites and lipids are extracted independently, using two replicates of the same sample: one for metabolomics and one for lipidomics [12,13,14,15]. This doubles the number of samples that need to be generated and analyzed, which is sometimes difficult to implement when the biological material is limited (e.g., cellular, other in vitro models, or precious tissues). Moreover, and this is a critical concern, performing the analysis of polar metabolites and lipids on distinct biological samples complicates further statistical integration and aggregation of data [16]. To overcome this limitation, new methods have been developed, with the aim to perform metabolomics and lipidomics on one single sample, using a dual extraction. Most of these studies apply untargeted metabolomics and lipidomics [17,18,19,20,21,22], with the main objective to perform an extensive analysis of all measurable molecules in a sample, including unknown analytes [23]. Although, in theory, nontargeted approaches allow a wide and unbiased detection of metabolic changes, in practice, results’ interpretation is often limited by the small number of successfully identified metabolites. In addition, the large metabolome coverage enabled by untargeted approaches is counterbalanced by the fact that these approaches have a lower resolution power and do not allow quantification, therefore, offering limited accuracy for investigating specific metabolic pathways. A few methods, such as the SIMPLEX method, which combines untargeted and targeted approaches, have been developed to enable the detection of several lipids and metabolites classes [24,25]. Being able to accurately and quantitatively measure changes in metabolites involved in specific key pathways is essential to go beyond the identification of metabolic biomarkers and to carry out fluxomics and isotope tracing experiments to progress toward the functional analysis of the metabolism. This requires the use of targeted approaches with sufficient resolution power and accuracy [23,24,25,26]. Accurate quantification of metabolites is especially important with respect to a precise characterization and understanding of the key biosynthesis and regulation pathways, such as central carbon energetic pathways [27]. Because these pathways include many isobaric metabolites (isocitrate and citrate, hexose phosphate...), it is important to develop strategies with high resolutive techniques and adequate chromatographic systems to be able to separate them.

Another critical point for having accurate quantitative data is to reduce the variability introduced during the experimental protocol. Ahead of the analytical steps, metabolomics and lipidomics approaches require many different steps, including sample preparation and cell and metabolite extraction, which often generate a large variability in the measurements and may limit the precision and reliability of these approaches. Labeled internal standards can be used to improve the accuracy of the quantification, as proposed in a few studies [25] Indeed, the application of the Isotope Dilution Mass Spectrometry (IDMS) approach, which is based on the use of ^13^C-chemical labeled internal standards, considerably reduces errors caused by variations occurring during analysis and sample processing [28,29,30], as for instance, inter-operator variability in the different sample processing steps, or the ionization loss effects in mass spectrometry, which impact analytical precision [31].

Sample collection is also a crucial step to ultimately achieve the unbiased monitoring of a maximum of compounds and obtain a faithful picture of the metabolic state. This step encompasses different stages of the dedicated protocol and needs to be adapted to the model used for the study (blood, tissues, urine, feces, cells…) [32]. For all matrices, sample collection has to be performed as fast as possible to immediately block all enzymatic processes and prevent nonphysiological modifications of metabolites. This process is called quenching [33,34]. Incorrect or non-suitable quenching protocols can result in an alteration of the level of some polar metabolites, since many are extremely labile, especially metabolites in central carbon metabolism pathways, which can be rapidly modified in response to changes induced by the sampling. Thus, this step is essential to gain an accurate overview of the real concentrations of these metabolites [35]. There are multiple different ways to stop the metabolic activity of adherent cells. Currently, the most often used method consists of a liquid extraction using different organic solvents, which are usually adapted for specific metabolite extractions [36]. Several studies have shown that each extraction solvent provides advantages and disadvantages for the recovery of different types of metabolites [32,37]. A fast-sampling method was developed by Martano et al. [34] consisting of a fast and efficient washing of cells harvested on a coverglass, followed by a fast and reproducible metabolism quenching of enzymatic processes. This method appears to be particularly suited for qualitative and quantitative assessment of the metabolism of adherent cells, providing more reproducible and less biased results.

In this study, we propose a method that combines the fast-sampling method developed by Martano et al. [34], with a double extraction method adapted from the classical liquid–liquid extraction used for lipids. The method uses ^13^C labeled internal standards and different analytical systems (ion chromatography and pentafluorophenyl, PFP, separation for polar metabolites) to reach the best possible resolution and qualitative and quantitative accuracy of measurements. This especially allows getting an extensive coverage and accurate quantification of the metabolites involved in the central carbon energy pathways (glycolysis, pentose phosphate, and Tricarboxylic Acid (TCA) cycle pathways). The aim is to provide an accurate and faithful snapshot of the metabolism using a single sample, by both maximizing the detection of polar metabolites and lipids within the same sample and simultaneously minimizing as much as possible unwanted experimental (biological and technical) variations.

Being able to accurately and reliably measure the amounts and changes in metabolites and lipids is of particular interest for investigating subtle metabolic perturbations, as those occurring in the context of chronic exposure to food or environmental contaminants. We, therefore, applied our method to HepaRG cells, a cell line derived from a human hepatocellular carcinoma [38], and which is increasingly used in toxicology studies. This cell line has the particularity of differentiating from bipotent progenitor cells into mature hepatocyte-like cells. The fully differentiated HepaRG cells have been shown to display metabolic capacities close to primary human hepatocytes, making them a good cell model for studying liver metabolism. Among others, this cell line is implemented in large EU H2020 projects as well as in the US Tox21 program [39,40], both of which imply the testing of a large range of chemicals hypothesized to impact human metabolism. Besides being largely used for toxicological studies and displaying many advantages regarding, for instance, their closeness to primary human hepatocytes in terms of metabolic capacity, HepaRG cells are quite complex to grow, requiring a long period to differentiate, which often limits the number of available samples. It is, therefore, all the more of interest to be able to apply on those cells a double extraction protocol, which allows the simultaneous study of the metabolome and lipidome while requiring half the samples. Then, this new protocol was applied to explore the metabolic effects of fenofibrate, a drug known to impact liver metabolism and used as a model compound in hepatotoxicity studies. Although fenofibrate is suspected to affect different metabolic pathways, most studies have focused on its effects on lipid metabolism. We used our new double extraction method to explore broader metabolic effects of fenofibrate in the human liver, using the HepaRG cell line.

## 2. Results and Discussion

### 2.1. Qualitative and Quantitative Analysis of the Cell Lipidome and Metabolome Using a Newly Developed Dual Extraction Protocol

The main objective of this study was to develop a new method, further referred here as the “double extraction protocol”, and assess its interest. This protocol aims to perform metabolomics and lipidomics analyses on one unique biological sample with the objective of reducing the number of samples required for this type of analysis. It also allows an absolute quantification of an extended range of polar metabolites to be gained to explore changes more accurately in specific metabolic pathways. We compared this new protocol to a classically used method, which requires two different samples to measure polar metabolites and lipids. Briefly, in the classical extraction protocol (Figure 1A), polar metabolites for metabolomics analyses were extracted from a first sample using a specific quenching solution in the presence of fully ^13^C-labeled Escherichia coli (*E. coli*) extract as internal standard and obtained as previously described [28]. The solution containing extracted polar metabolites was then evaporated and analyzed. Lipids were extracted from a second sample using a biphasic extraction in the presence of specific internal standards (non-natural fatty acid chains) for each lipid family. Two phases were obtained, and the organic phase from which lipids was recovered was concentrated and analyzed. In the double extraction procedure (Figure 1B), we used a liquid–liquid extraction protocol combined with a specific quenching solution to stop the metabolism and prevent the rapid degradation and enzymatic modification of some polar metabolites. Then polar metabolites and lipids were extracted from the same sample using a biphasic extraction. The aqueous phase and the organic phase were used for metabolomics and lipidomics analyses, respectively, with the same internal standards used for the classical extraction. Fully ^13^C-labeled *E. coli* extract was added to the quenching solution in both protocols, to enable Isotope Dilution Mass Spectrometry (IDMS) [28] for the absolute quantification of polar metabolites.

We applied both methods to study the metabolome and lipidome of HepaRG cells, which is derived from a human hepatocellular carcinoma and largely used in toxicology for the study of potential hepatotoxicants and hepatic metabolism modulators. We compared these two protocols for their efficiency to extract some targeted polar metabolites and lipids species. Targeted metabolomics and lipidomics were performed for 40 polar metabolites and 82 lipids, using different analytic systems. For lipidomics analysis, two analytical systems were necessary to ensure optimal coverage of all the lipid classes studied: hydrophilic interaction liquid chromatography (HILIC) for phospholipids and sphingolipids and gas chromatography for neutral lipids. For polar metabolites, two distinct chromatographic methods were also used: reversed-phase to measure amino acids and ion chromatography to measure energy metabolites. With this strategy, we obtained the most extended and accurate coverage of all lipids and polar metabolites particularly for metabolites belonging to the carbon and energetic central metabolism (glycolysis, TCA cycle, and pentose phosphate pathways), which include many isobaric metabolites (isocitrate and citrate, hexose phosphate...). Absolute quantification (in nM) was obtained for 32 polar metabolites using the IDMS approach and calibration curves of corresponding chemical standards. Relative abundance results (e.g., the ratio with corresponding fully ^13^C-labeled compounds) were obtained for 8 polar metabolites and all the 82 lipids, by using ratios between area of the molecule of interest and their internal standards. Polar metabolites and lipids analyses were implemented in two separate laboratories, each of them having the required equipment to perform these analyses.

### 2.2. Polar Metabolites

For the majority of polar metabolites, the measured absolute/relative concentrations were similar for the two protocols. Only a few polar metabolites were found to be extracted in smaller amounts when using the double extraction protocol. These were the amino-acids phenylalanine and methionine (Figure 2A,B), as well as some energy metabolism linked metabolites: 6-phosphogluconate, a-ketoglutarate, cytidine diphosphate, guanosine diphosphate, pyridoxal-5-phosphate, uridine diphosphate acetylglucosamine and uridine 5′-monophosphate (Figure 2C,D).

For these polar metabolites, the addition of dichloromethane in the double extraction protocol, which results in the formation of two phases, may explain these differences. Indeed, the polarity of metabolites is an important property with regard to their distribution during liquid–liquid extraction [41]. Polar metabolites have more affinity for aqueous phases, while apolar metabolites preferentially migrate into organic phases [42,43]. Metabolites of intermediate polarity are often distributed among each phase. Therefore, when their concentration is assessed in the aqueous phase, it can be underestimated. Several concentrations measured using the double extraction method were found to be slightly lower for polar metabolites. However, these differences remained very minor among the two protocols. Still, for further analyses, special attention should be paid to polar metabolites as their absolute concentration and relative abundance may be underestimated. For these reasons, the qualitative distribution of all polar metabolites was systematically compared for the two protocols. To this end, the respective percentage of every individual amino acid, with respect to the total amino acid pool, as well as every energy metabolite, with respect to the total energy metabolite pool, was calculated to enable consideration of its relative distribution in its respective class. As shown in Figure 3, no significant difference was observed between the relative distributions obtained by the two protocols, with regard to both classes of compounds. Therefore, the relative distribution of polar metabolites was not impacted by the extraction protocol (Appendix A further details these results for low percentage species).

### 2.3. Apolar Metabolites

With regard to lipids recovery: phospholipids (phosphatidylcholines (PC), phosphatidylethanolamines (PE), phosphatidylinositols (PI)), sphingolipids (Ceramides (Cer), and sphingomyelins (SM)), cholesterol and triglycerides (TG), respectively, were relatively quantified using their corresponding internal standard as described in the Material and Methods section. For each of these lipid classes, the relative abundance of each of the molecular species was summed to obtain the total relative abundance of the class (Figure 4). The double extraction protocol was shown to provide a significantly better extraction for cholesterol and ceramides. Conversely, a significantly better extraction was observed for the classical extraction procedure for TG, PC, PE, and PI. SM were similarly extracted by both protocols.

Although some lipid classes were found to be extracted more efficiently using one of the two methods, the qualitative profiling, i.e., the relative distribution of lipid species in each family, was not impacted by the extraction protocol. Indeed, for all lipid classes, there was no difference in the percentage of the different molecular species (Figure 5 for TG and SM; Appendix A for Cer, PC, PE, and PI).

### 2.4. Methods Reproducibility

To evaluate the reproducibility of both methods, coefficients of variation (CV) were calculated for all individual polar metabolites and lipids. Results showed a significantly improved reproducibility for the double extraction method, compared to the classic protocol, for amino acids (CV_Double extraction_ = 17.38 ± 17.04%; CV_Classical extraction_ = 36.10 ± 20.68%), Cer (CV_Double extraction_ = 6.35 ± 1.80%; CV_Classical extraction_ = 9.51 ± 1.71%), and PI (CV_Double extraction_ = 4.73 ± 1.88%; CV_Classical extraction_ = 10.95 ± 1.42%), respectively. Conversely, variability was found to be slightly higher with the double extraction method for SM (CV_Double extraction_ = 13.11 ± 2.47%; CV_Classical extraction_ = 9.99 ± 3.05%), although it remained within an acceptable range. No difference was observed for energy metabolites, TG, PC, and PE, respectively. All graphs comparing CVs for polar metabolites and lipids are provided as Appendix A.

Possibly, the use of acetonitrile in the double extraction protocol could minimize the degradation of some polar metabolites as well as lipids during the extraction process [44], therefore, explaining the improvement in some CVs for the double extraction method and differences in lipids classes recovery in Figure 4.

Taken together, these results showed that all targeted polar metabolites and lipids could be efficiently detected using both extraction procedures, and thus that the double extraction protocol was able to detect and quantify all classically measured amino acids, energy metabolites, and lipids. Although few polar metabolites and lipid species were found to be better extracted with the classical extraction protocol, on quantitative bases, the relative distribution of polar metabolites and lipid species within each family was demonstrated to be identical (Figure 3 and Figure 5).

Despite these slight quantitative differences, the double extraction protocol provides many advantages. This method was demonstrated to allow a similar coverage of the metabolome as well as the lipidome, compared to a classical extraction, while relying on the use of one single sample. This is of particular interest when the amount of biological material is limited, which is the case for many types of samples, including biopsies for cancer detection, for instance [45], certain types of cells, such as stem cells [46], or cerebrospinal fluid [47]. In addition, only one extraction is required for the double extraction protocol, which considerably reduces the experimental time and cost. Being able to quantify metabolites and lipids out of the same sample also facilitates the aggregation of data for robust statistical analyses, and, therefore, opens the road for more accurate biological interpretation, in particular, by limiting inter-sample variability. One counterpart of this protocol, which enables broad coverage of the metabolome, is that its implementation requires larger analytical resources and access to advanced material and facilities, which are not available in all laboratories. However, such advanced protocols are particularly interesting for current and future studies aiming to explore the metabolic effects of chemicals, whether they involve the direct use of human samples, or based on human cellular models. The latter approaches are increasingly employed to assess the toxicity of chemical pollutants and drugs, using high throughput approaches in which model toxicants (such as fenofibrate for the liver) are used in parallel to large sets of test molecules, to understand the mode of action of these toxicants better.

### 2.5. Exploration of the Metabolic Effects of Fenofibrate Using the Double Extraction Protocol

The double extraction protocol was used to study the effect of fenofibrate on liver metabolism. To do so, HepaRG cells (n = 18) were exposed to a high concentration (450 µM) of fenofibrate solubilized in DMSO (0.25%). The main objective of this experiment, carried out at a high concentration, was to validate the ability of our double extraction protocol to highlight modulation of polar metabolites and lipids on the same sample, and assess whether this protocol is suitable to study the metabolic effects of a compound, with a large coverage of the metabolome. Fenofibrate is a PPARα agonist and is clinically used as a lipid-lowering agent to treat hypertriglyceridemia and hypercholesterolemia [48]. Numerous studies have been published focusing on the effects of fenofibrate on lipid metabolism in rodents, especially [49,50]. However, species differences have been demonstrated in response to PPARα agonists, particularly as it refers to the differences between humans and rodents [51]. In humans, most studies have explored the effects of fenofibrate in plasma [52], showing that an activation of PPARα by fenofibrate results in a decrease in plasma triglycerides and LDL cholesterol concentrations and an increase in HDL cholesterol [53]. Although few studies have looked at the effect of fenofibrate on the human liver, fenofibrate is suspected to influence hepatic lipid homeostasis and energy balance. In line with this hypothesis, our results showed that 46 polar metabolites and lipids were affected by fenofibrate (Figure 6). The results for all measured polar metabolites and lipids are provided in Appendix A.

#### 2.5.1. Effects on Lipids

Regarding lipids, a majority of Cer and SM was found to be reduced. Conversely, all measured TG species were increased by fenofibrate. Few PI and PE were altered by fenofibrate with either a decrease or increase in their concentration depending on the species. No significant differences were observed for PC.

In line with our reported increase in liver TG after exposure to a high concentration of fenofibrate, previous studies have similarly demonstrated TG accumulation in the liver in rodents and HepG2 cells (human hepatocytes) exposed to fenofibrate [49,54]. However, the effects of fenofibrate on the liver TG remain controversial, with other studies performed on HepaRG cells and rodents, showing that fenofibrate either lowers TG accumulation [55,56,57] or does not affect hepatic TG content [58]. These different results may be explained by the use of different cellular models. Regarding Cer and SM, this is the first report showing that these compounds decrease in liver cells following fenofibrate exposure. It was previously reported that the SM and Cer content of hepatic cells exposed to fenofibrate was not impacted by exposure, but this was at low concentrations only [58]. Conversely, an in vivo study has reported a decrease in both these compounds in human plasma [52]. The authors observed a correlation between these two families, suggesting that Cer and SM could be similarly impacted by fenofibrate treatment, which is in line with the results of the present study, where we also observed that these two lipids families were similarly decreased by fenofibrate.

#### 2.5.2. Effects on Polar Metabolites

Regarding polar metabolites, the concentration of some amino acids, mainly aliphatic amino acids (glycine, proline, leucine, isoleucine, valine), but also methionine and histidine, was significantly affected by exposure to fenofibrate. All of them had higher concentrations in HepaRG exposed cells compared to controls.

Fenofibrate was reported to similarly affect amino acid levels in mice plasma, including glycine, isoleucine, leucine, valine but also glutamate, phenylalanine, serine, tryptophan, and tyrosine [50]. Although these results are not directly comparable with ours, as the matrices (intra- or extrahepatic content) are different, they strongly suggest that fenofibrate impacts the hepatic metabolism of amino acids.

Many energy metabolites were also found to be impacted by fenofibrate, including metabolites from the glycolysis and pentose phosphate pathway (G6P, Fru1P, phosphoenolpyruvate (PEP), and Sed-7P), and TCA cycle (citrate). All of them had decreased concentrations in exposed cells, except for P-ser and Sed7P, whose concentrations were significantly increased in exposed cells.

Our observations suggest that G6P and Sed7P, both belonging to the pentose phosphate pathway, were, respectively, decreased and increased by fenofibrate. These observations are in accordance with the results reported by Oosterveer et al. [59], who showed that hepatic G6P content was reduced in the liver of fenofibrate-treated mice. These authors concluded that the pentose phosphate pathway was presumably enhanced by fenofibrate.

In the present study, we also observed a decrease in citrate and PEP, two metabolites belonging respectively to the TCA cycle, and to the last step of glycolysis. In addition, ATP, which is produced by the TCA cycle and glycolysis, was also found to be reduced. This point was already reported by Ohta et al. [60], with a reduction in TCA cycle intermediates, suggesting that energy metabolism homeostasis could be altered.

The fenofibrate concentration used in this study (450 µM) is higher than the therapeutic concentrations used in humans. Under therapeutic conditions for humans, a daily dosage of 145 mg of fenofibrate results in a maximal plasma concentration of 11.016 mg/L [61], which represents approximately 30 µM in blood. Therefore, this study does not allow the effect of fenofibrate at therapeutic doses to be directly concluded but could help to understand the effect of this molecule in the liver at higher and possibly toxic concentrations.

In this study, we used fenofibrate as a model molecule for liver toxicity. Based on our newly developed extraction procedure, which allows the quantitative and qualitative simultaneous profiling of polar and apolar metabolites, we were able to highlight a major modulation of the hepatic metabolome in HepaRG cells. We confirmed that fenofibrate impacts hepatic lipid metabolism in human cells, but also has broader effects on liver metabolism, modulating many polar metabolites involved in different metabolic pathways.

Hepatic metabolism is nowadays suspected to be impacted as well by many man-made chemicals to which humans are unintentionally exposed. These “Metabolism Disrupting Chemicals” are among the chemical pollutants which raise serious concerns in environmental and food toxicology [62]. In this study, we demonstrated that our double extraction protocol is particularly suited to detect metabolic modulations spanning different metabolic pathways accurately, and, therefore, could be efficiently used in the future to decipher the metabolic effects induced by such food and environmental contaminants, for large sets of molecules. This is the case not only for the liver but also for other tissues and biofluids, especially when the amount of biological material is limited.

## 3. Materials and Methods

### 3.1. Reagents

DMSO (dimethyl sulfoxide), penicillin, streptomycin, trypsin, fenofibrate, and PBS were obtained from Sigma–Aldrich (St. Quentin Fallavier, France). The concentration of the stock solutions was 10 mM in DMSO. Solvent for the extraction solution (methanol, acetonitrile, formic acid, dimethyl sulfoxide (DMSO)) were obtained from ThermoFisher Scientific (Illkirch, France). Methanol (MeOH), dichloromethane (CH_2_Cl_2_), ethyl acetate (EtAc), ammonium acetate (AmAc, chemical purity >99%), ethylene glycol-bis (2-aminoethlether)-N,N,N’,N’-tetraacetic acid (EGTA with chemical purity >97%), boron trifluoride-methanol (BF3-MeOH 10% w/w), hexane (Hex) were purchased from Merck (Fontenay sous Bois, France). Acetonitrile (ACN) was provided by ThermoFisher Scientific and water Milli-Q (H_2_O) by Millipore. Lipid standards: ceramide d18:1/15:0 (Cer); phosphatidylethanolamine 12:0/12:0 (PE); phosphocholine 13:0/13:0 (PC); sphingomyelin d18:1/12:0 (SM); phosphoinositol 15:0/18:1-d7 (PI); phosphoserine 12:0/12:0 (PS) were purchased from Avantis Polar Lipids. Stigmasterol (STIG) for cholesterol and triglyceride-19 (TG) were purchased from Merck.

### 3.2. Cell Culture and Treatment

The HepaRG cells, kindly given by Dr. C. Guguen-Guillouzo, were cultured according to the standard protocol initially described by Gripon et al. [38]. Cells were cultivated in a humidified atmosphere of 5% CO_2_ at 37 °C for two weeks in phenol red William’s Eagle Medium supplemented (ThermoFisher Scientific) with 10% FCS *v*/*v* (PAN biotech, Dutscher, Brumath, France), 100 units/mL penicillin, 100 μg/mL streptomycin, 5 μg/mL insulin 200 mM glutamine, and 5 × 10^−5^ M hydrocortisone hemisuccinate. The cells were grown in the same medium, with 2% DMSO for two additional weeks (cell differentiation). Differentiated cells were then harvested using trypsin and were plated on a 30 mm-coverglass (Dutscher) in 6-well plates (Dutscher) at a density of 500,000 cells/well in 4 mL medium per well. Cells were washed in PBS 24 h after seeding, and the medium was replaced by 2 mL of phenol red-free William’s Eagle Medium supplemented with 5% FCS *v*/*v* and with the same additives as described above. DMSO was used as a vehicle (negative control) and added to the plates with a final DMSO concentration of 0.25% in the culture medium. In a first experiment, cells were cultured during 24 h in the phenol red-free William’s Eagle Medium and extracted with 2 different extraction protocols (classical extraction and double extraction) as described above (n = 5 replicates per condition). In a second experiment, cells were exposed during 24 h to either a high concentration of fenofibrate (450 µM) or vehicle only and were extracted using the double extraction protocol (n = 18 replicates per condition).

### 3.3. Sampling

The sampling protocol used in this study is adapted from a fast-sampling method for metabolites extraction published by Martano et al. [34]. Briefly, a coverglass was placed into each well of the six-well plates, and cells were seeded on top. At the end of the treatment period, the six-well plates with adherent cells on the coverglass were placed on a heating block at 37 °C for mammalian cells. Each coverglass was carefully grabbed and washed in a stirring Milli-Q water bath and then dropped into a cold 4-mL quenching solution containing internal standards with the cells facing upward. A mix of cold (−20 °C) acetonitrile, methanol, and Milli-Q water with 0.1% formic acid 2:2:1 *v*/*v* was used as a quenching solution. Several studies have demonstrated that the use of a mixture of solvents is optimal for maximizing the number of recovered polar metabolites from adherent cells, and acidified Milli-Q water with formic acid and cold (−20 °C) solvents have been shown to stabilize various phosphorylated compounds, including phosphorylated sugar, nucleotides or phosphorylated fatty acids, and heat-sensitive metabolites. Then, a cell scraper was used to recover the cells from the coverglass, and the cell suspension was pipetted into a 15 mL centrifugation tube. Finally, the suspension was sonicated for 30 s and subsequently incubated for 15 min on ice, followed by 30 min in liquid nitrogen. Samples were then stored at −80 °C until the extraction.

### 3.4. Classical Extraction Method

In the classical extraction procedure, polar metabolites and lipids were obtained on two independent samples. Polar metabolites were extracted using the quenching solution described above consisting of a mix of cold (−20 °C) acetonitrile, methanol, and milliQ water with 0.1% formic acid 2:2:1 *v*/*v*. Two fully ^13^C-labeled extracts from *E. coli* were used for the IDMS approach: one from proteinogenic amino acids extraction for amino acids quantification and one form intracellular extraction for central metabolites. Fifty microliters of ^13^C labeled proteinogenic amino acids and 50 µL of ^13^C labeled intracellular extract were added as internal standard. Each sample was evaporated to dryness using a speed-vacuum (Thermo scientific), resuspended in 100 µL of milliQ water, and centrifuged at 400× *g* for injection. Lipids were obtained from the cell homogenized in using 2 mL of cold PBS, 75 µL of internal standard dissolved in methanol, containing Cer d18:1/15:0 16 ng; PE 12:0/12:0 180 ng; PC 13:0/13:0 16 ng; SM d18:1/12:0 16 ng; PI 15:0/18:1-d7 30 ng; PS 12:0/12:0 156.25 ng; stigmasterol 4 µg, and triglyceride-19 12 µg were added. Samples were first centrifuged at 400× *g*, and lipids were then extracted according to an adapted Bligh and Dyer extraction with an addition of 2.5 mL methanol and 2.5 mL dichloromethane. The aqueous phase (upper phase) was removed, and the organic phase (lower phase) was transferred to a glass tube, dried under nitrogen flow, transferred to a glass HPLC vial and resuspended with the appropriate solvent: 50 µL MeOH to analyze phospholipids by LC/MS/MS or 20 µL of ethyl acetate to analyze neutral lipids in GC/FID. In total, 5 replicates were carried out for each extraction type of this classical procedure, 5 replicates for polar metabolites, and 5 replicates for lipids.

### 3.5. Double Extraction Method

In the double extraction procedure (Figure 1B), polar metabolites and lipids were obtained on one unique sample. Polar metabolites and lipids were extracted using a quenching solution consisting of a mix of cold (−20 °C) acetonitrile, methanol, and milliQ water with 0.1% formic acid 2:2:1 (*v*/*v*/*v*). Internal standards were added for polar metabolites and lipids using the same volume as described for the classical extraction. Samples were centrifuged at 400× *g*, polar metabolites and lipids were separated using a Bligh and Dyer modified extraction with 2.5 mL of dichloromethane. The aqueous phase (upper phase) was kept, evaporated, and resuspended in 100 µL of milliQ water for metabolites analysis. The organic phase (lower phase) was treated in the same way as described for the classical extraction procedure for apolar metabolites.

### 3.6. Neutral Lipids Analysis

Extracted lipids were resuspended in 20 µL of ethyl acetate, and 1 µL was injected to analyze neutral lipids by gas–liquid chromatography on a Focus Thermo Electron system using a Zebron-1 Phenomenex fused-silica capillary column (5 m, 0.32 mm i.d., 0.50 mm film thickness). The oven temperature was programmed from 200 to 350 °C at a rate of 5 °C/min, where the carrier gas was hydrogen (0.5 bar). The injector and the detector were at 315 and 345 °C, respectively.

### 3.7. Phospholipids and Sphingolipids Analysis

Extracted lipids were resuspended in 50 µL of methanol, and phospholipids were analyzed by liquid chromatography, Infinity 1290 (Agilent Technologies, Les Ulis, France), coupled with a triple quadripole 6460 (Agilent Technologies, Les Ulis, France) equipped with electrospray ionization (ESI). Samples were analyzed in the positive for Cer, SM, PE, PC, and negative mode, for PI, respectively. The source parameters were source temperature 325 °C, nebulizer gas (nitrogen) flow rate 10 L·min^−1^, sheath gas (nitrogen) temperature 400 °C, sheath gas (nitrogen) flow rate 12 L·min^−1^, spray voltage 4000 V. The analysis was performed in selected reaction monitoring (SRM). A HILIC column (50 × 4.6 mm, 2.6 µm, Phenomenex), maintained at 40 °C was used, with a mobile phase A consisting of acetonitrile and a mobile phase B consisting of 10 mM ammonium acetate in water, pH 3.2. In positive mode, solvent B varied as follows: 0 min: 10%, 10 min: 30%, 11 min: 100%, 12 min: 100%, 13 min: 10%, 15 min: 10%. The flow rate was 300 µL·min^−1^, and the volume of the injection was 2 µL. In negative mode, solvent B varied as follows: 0 min: 5%, 10 min: 50%,11 min: 5%, 15 min: 5%. The flow rate was 800 µL·min^−1^, and the volume of the injection was 5 µL.

### 3.8. Energy Metabolites Analysis

Analysis of intracellular energy metabolites was performed by high-performance anion-exchange chromatography (Dionex ICS 5000 + system, Sunnyvale, CA, USA) coupled with a LTQ Orbitrap Velos mass spectrometer (Thermo Fisher Scientific, Waltham, MA, USA), equipped with a heated ESI probe. Samples were analyzed in the negative Fourier-transform mass spectrometry (FTMS) mode at a resolution of 60,000 (at *m*/*z* 400) with the following source parameters: capillary temperature 300 °C, source heater temperature 250 °C, sheath gas flow rate 30, auxiliary gas flow rate 10, S-Lens RF level 50%, and source voltage 2.5 kV. The injection volume was 15 µL. Samples were injected on a Dionex IonPac AS11 column (250 × 2 mm) equipped with a Dionex AG11 guard column (50 × 2 mm). The mobile phase was composed of a KOH gradient which varied as follows: 0 min 0.5; 1 min 0.5; 9.5 min 4.1; 14.6 min 4.1; 24 min 9.65; 36 min 60; 36.1 min 90; 43 min 90; 43.1 min 0.5; 45 min 0.5.

### 3.9. Amino Acids Analysis

Analysis of intracellular amino acids was performed as described previously in Heuillet et al. [63]. Briefly, the analysis was performed by liquid chromatography (HPLC U3000, Dionex, Sunnyvale, CA, USA) coupled with a Qexactive mass spectrometer (Thermo Fisher Scientific, Waltham, MA, USA) equipped with a heated ESI probe. MS analyses were performed in the positive FTMS mode at a resolution of 60,000 (at *m*/*z* 400) with the following source parameters: capillary temperature 275 °C, source heater temperature 250 °C, sheath gas flow rate 45, auxiliary gas flow rate 20, S-Lens RF level 40%, and source voltage 5 kV. Samples were injected on a Supelco HS F5 Discovery column (150 mm × 2.1 mm; 5 µm particle size) equipped with a Supelco HSF5 guard column (20 mm × 2.1 mm; 5 µm particle size). Solvent A was 0.1% formic acid in H2O, and solvent B was 0.1% formic acid in acetonitrile at a flow rate of 250 µL·min^−1^. Solvent B was varied as follows: 0 min: 2%, 2 min: 2%, 10 min: 5%, 16 min: 35%, 20 min: 100%, 24 min: 100%, 24.1 min: 2% and 30 min: 2%. The volume of the injection was 5 µL. Identification was determined by extracting the accurate mass of amino acids with a mass accuracy of 5 ppm.

### 3.10. Statistical Analysis

Results are expressed in relative abundance or in concentration (nM) for lipids and polar metabolites. For lipids, relative abundance was obtained using the ratio of a specific lipid species with its corresponding internal standard for each class of lipids. For polar metabolites, relative abundances were calculated from the ratio between the total signal arising from the metabolite and the isotopic standard (^12^C/^13^C) [27]. Absolute concentrations in nM were calculated from calibration curves constructed using standard solutions of (unlabeled) metabolites at different concentrations mixed with equal amounts of isotopic standards, as detailed in Wu et al. [28]. For each metabolite, two Student’s *t*-tests were performed to compare (1) the two extraction procedures (dual extraction vs. classical extraction) and to assess the effect of fenofibrate (fenofibrate-treated cells vs. non-treated cells). The average CVs obtained for polar metabolites and all classes of lipids for the two extraction methods were also compared using Student’s *t*-tests to test the difference in variability between the two methods. For lipid classes, the relative abundance of lipid species within each class (expressed as a percentage of the total class) was compared for the two extraction procedures and for the fenofibrate-treated vs. non-treated cells using the Khi2 test. Differences were considered significant when p-value *p* < 0.05. Statistical analyses were performed using R software. Data were represented as mean ± standard deviation with Graphpad Prism 8.4.1 software. All raw data have been uploaded to the Metabolights repository (URL http://www.ebi.ac.uk/metabolights/MTBLS1835).

## 4. Conclusions

In this study, we showed evidence that a double extraction protocol performed on a single sample can provide excellent profiling of both polar and apolar metabolites, with similar results compared to a classical extraction performed on two different samples. This new protocol has been validated on HepaRG cells, a pivotal cell line increasingly used in toxicology for the study of potential hepatotoxicants and hepatic metabolism modulators. This newly developed protocol allowed us to reach very satisfactory quantitative as well as qualitative profiling. Applied to hepatic cells treated with the model molecule fenofibrate, the double extraction protocol enabled showing that many metabolites were impacted by exposure, demonstrating its capacity to highlight marked modulation of both polar metabolites and lipids simultaneously. In metabolomics, sample preparation is always a critical point. In this respect, this double extraction protocol provides many advantages in terms of experimental and analytical time, reduces total costs, and can easily be applied to many different models, particularly when the number of biological samples is limited. The analytical strategy implemented in this method enables reaching a high resolutive power for key metabolic pathways and, especially, to get an extended and accurate coverage of the carbon and energetic central metabolism (i.e., glycolysis, TCA cycle, and phosphate pentose pathways), which will enhance the quality of future studies devoted to the functional analysis of cellular metabolism using isotope profiling and fluxomic approaches. Although this advanced protocol requires access to facilities with the appropriate technologies, it will be highly beneficial for various projects using global and semi-targeted metabolomics and lipidomics with the aim to explore metabolic modulations, such as the ones suspected for many food or environmental contaminants, and which testing implies the need to assess numerous molecules, using high-throughput compatible approaches.

## Figures and Tables

**Figure 1 metabolites-10-00338-f001:**
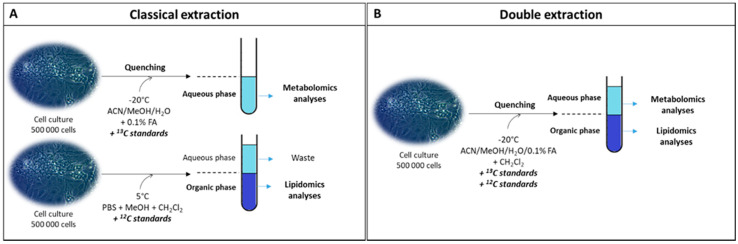
Experimental scheme of the classical and double extraction methods used to perform metabolomics and lipidomics analyses. For the classical extraction (**A**), a mix of acetonitrile (ACN), methanol (MeOH), and acidified water (H_2_O) with formic acid (FA) was used to perform polar metabolites’ extraction. Phosphate buffer saline (PBS) associated with dichloromethane (CH_2_Cl_2_) and methanol was used to extract lipids. For the double extraction method (**B**), the mix of acetonitrile, methanol, and acidified water was supplemented with dichloromethane to extract polar metabolites and lipids on the same sample. ^13^C and lipids internal standards were added to perform absolute quantification and relative quantification for metabolomics and lipidomics analysis, respectively.

**Figure 2 metabolites-10-00338-f002:**
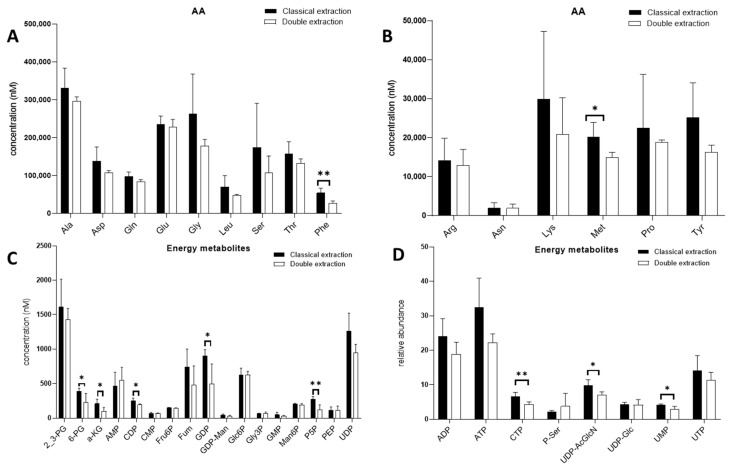
Comparison of polar metabolites quantitative recovery for the classical (black bars) and double extraction (white bars) methods. Amino acids (**AA**, **A**,**B**) and energy metabolites (**C**,**D**) were quantified using ^12^C/^13^C ratio for each metabolite. Concentrations (**A**–**C**) were expressed in nM using an external calibration. Relative abundance (**D**) was obtained using only ^12^C/^13^C ratios. Bars represent mean ± SD (n = 5). *t*-tests were performed for each metabolite (* *p* < 0.05; ** *p* < 0.01). alanine (Ala), asparagine (Asp), glutamine (Gln), glutamate (Glu), glycine (Gly), leucine (Leu), serine (Ser), threonine (Thr), phenylalanine (Phe), arginine (Arg), asparagine (Asn), lysine (Lys), methionine (Met), proline (Pro), tyrosine (Tyr), 2,3-bisphosphoglycerate (2_3PG), 6-phosphogluconate (6-PG), a-ketoglutarate (a-KG), adenosine 5′-monophosphate (AMP), cytidine diphosphate (CDP), cytidine 5′-monophosphate (CMP), fructose-6-phosphate (Fru6P), fumarate (Fum), guanosine diphosphate (GDP), guanosine diphosphate mannose (GDP-Man), glucose-6-phosphate (Glc6P), glycerol-3-phosphate (Gly3P), guanosine 5′-monophosphate (GMP); mannose-6-phosphate (Man6P), pyridoxal-5-phosphate (P5P), phosphoenolpyruvate (PEP), uridine diphosphate (UDP), adenosine diphosphate (ADP), adenosine triphosphate (ATP), cytidine triphosphate (CTP), phosphoserine (P-Ser), uridine diphosphate acetylglucosamine (UDP-AcGlcN), uridine diphosphate glucose (UDP-Glc), uridine 5′-monophosphate (UMP), uridine triphosphate (UTP).

**Figure 3 metabolites-10-00338-f003:**
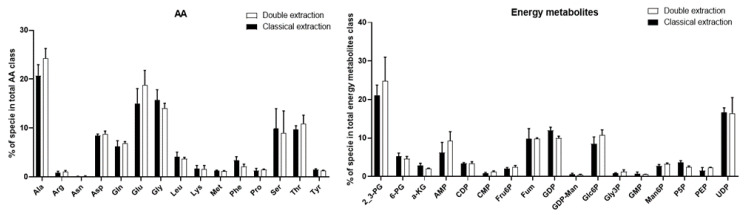
Comparison of the qualitative distribution of amino acids (**AA**) and energy metabolites species for the classical and double extraction methods. Total polar metabolites classes were considered as 100%. Each species corresponds to the percentage of its total polar metabolites classes. Bars represent mean ± SD (n = 5). A Khi2 (χ2) test was performed for each class to compare the relative distribution between both methods. No significant differences were observed for alanine (Ala), asparagine (Asp), glutamine (Gln), glutamate (Glu), glycine (Gly), leucine (Leu), serine (Ser), threonine (Thr), phenylalanine (Phe), arginine (Arg), asparagine (Asn), lysine (Lys), methionine (Met), proline (Pro), tyrosine (Tyr), 2,3-bisphosphoglycerate (2_3PG), 6-phosphogluconate (6-PG), a-ketoglutarate (a-KG), adenosine 5′-monophosphate (AMP), cytidine diphosphate (CDP), cytidine 5′-monophosphate (CMP), fructose-6-phosphate (Fru6P), fumarate (Fum), guanosine diphosphate (GDP), guanosine diphosphate mannose (GDP-Man), glucose-6-phosphate (Glc6P), glycerol-3-Phosphate (Gly3P), guanosine 5′-Monophosphate (GMP); mannose-6-phosphate (Man6P), pyridoxal-5-phosphate (P5P), phosphoenolpyruvate (PEP), and uridine diphosphate (UDP).

**Figure 4 metabolites-10-00338-f004:**
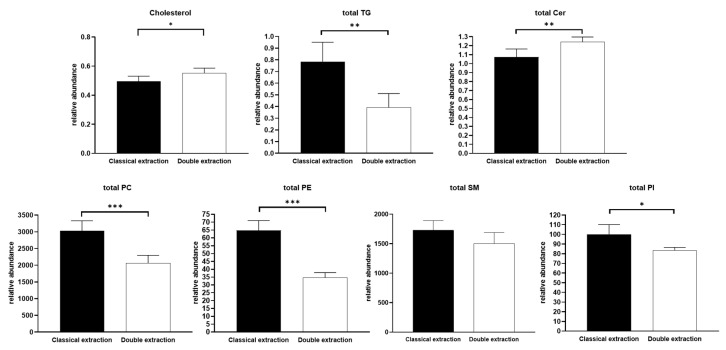
Comparison of lipid classes quantitative recovery between classical and double extraction methods. Total lipids classes correspond to an addition of all molecular species (ratio with their internal standard) obtained for each class. Bars represent mean ± SD (n = 5). *t*-tests were performed for each metabolite (* *p* < 0.05; ** *p* < 0.01; *** *p* < 0.001). Triglycerides (TG), ceramides (Cer), phosphatidylcholines (PC), phosphatidylethanolamines (PE), sphingomyelins (SM), phosphatidylinositols (PI).

**Figure 5 metabolites-10-00338-f005:**
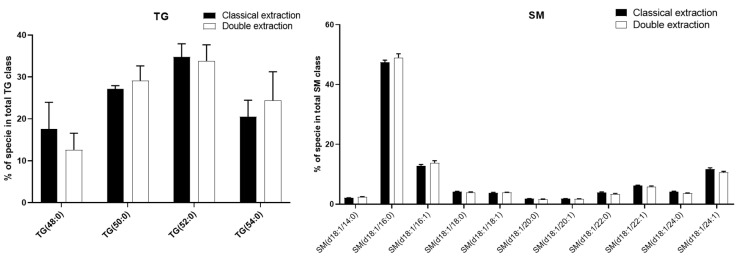
Comparison of the qualitative distribution of sphingomyelin (SM) and triglyceride (TG) species between the classical and double extraction methods. Each species is represented as a proportion (%) of its total class. Total lipids classes are considered as 100%. Bars represent mean ± SD (n = 5). A Khi2 (χ2) test was performed separately for each class to compare the relative distribution between both methods. No significant differences were observed.

**Figure 6 metabolites-10-00338-f006:**
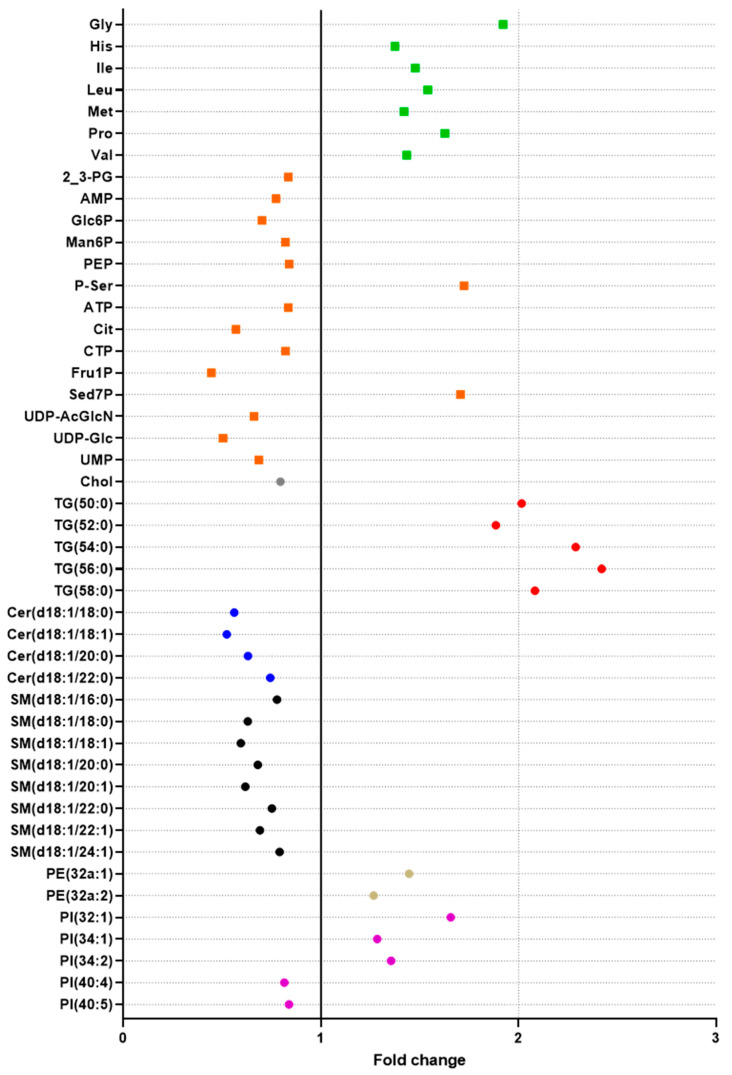
Effect of a high concentration of fenofibrate (450 µM) on HepaRG cells. Data were obtained with 18 replicates. *t*-tests were performed for each metabolite to compare non treated (DMSO) and treated (fenofibrate) cells (see Appendix A). All significant polar metabolites (◼) and lipids (⬤) are represented in this figure as fold change. Different colors were used to represent Amino acids (green), energy metabolites (orange), Cholesterol (grey), TG (red), Ceramides (blue), sphingomyelin (black), PE (brown), and PI (purple). Fold changes were obtained by dividing the means obtained for fenofibrate treatment by the means obtained for DMSO exposure. Increased metabolites in fenofibrate conditions are shown on the right (fold change > 1), and decreased metabolites in fenofibrate conditions are shown on the left (fold change < 1). glycine (Gly), phosphoserine (P-Ser), sedoheptulose-7-phosphate (Sed7P), proline (Pro), leucine (Leu), isoleucine (Ile), valine (Val), methionine (Met), histidine (His), phosphoenolpyruvate (PEP), adenosine triphosphate (ATP), 2,3-bisphosphoglycerate (2_3PG), cytidine triphosphate (CTP), mannose-6-phosphate (Man6P), cholesterol (Chol), adenosine 5′-monophosphate (AMP), glucose-6-phosphate (Glc6P), uridine 5′-monophosphate (UMP), uridine diphosphate acetylglucosamine (UDP-AcGlcN), citrate (Cit), uridine diphosphate glucose (UDP-Glc), fructose-1-phosphate (Fru1P), triglycerides (TG), ceramides (Cer), phosphatidylethanolamine (PE), sphingomyelins (SM), phosphatidylinositols (PI).

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
