# Peer review of "An Optimized Dual Extraction Method for the Simultaneous and Accurate Analysis of Polar Metabolites and Lipids Carried out on Single Biological Samples"

_metabolites, 2020, doi:10.3390/metabo10090338_

Round 1
Reviewer 1 Report
The author proposes new “double-extraction method” which the author claim as novel and better compared to the classical extraction method with a limited sample size.
Although the author has done a good job on the experimental part regarding comparison and quantification of the few metabolites and lipids. However, the following concerns make the overall manuscript and its claims questionable:
- The reported the “double extraction” method is not novel and already reported for simultaneous extraction of metabolite, lipids, and even protein in earlier reports. Also, the results indicated that it is comparable but not better than classical method of metabolite and lipid extraction.
- As indicated in the title “increased metabolome coverage” which is not clearly justified from the results author shown on fenofibrate study.
- The claim about “limited sample size”: the author used 0.5 million cells (Figure 1) which is a large sample size. The author should show that method can work in a few thousand cells compared to the classical method.
Considering these major drawbacks the study in the present form may not suitable for the publication. However, the analytical results from the study are well performed and can be reported along with raw data and protocols in a suitable publication platform or data repository.
Reviewer 2 Report
In this work, Villaret-Cazadamont et al. report a novel dual extraction to analyze metabolites and lipids in the same biological sample, useful in cases in which there is a size limitation of samples, avoiding the analysis of two different samples. The authors clearly demonstrate its applicability in toxicological studies to detect slight differences in metabolites and lipids, using HepaRG cells treated with fenofibrate.
The authors claim that this new protocol is suitable to analyze samples involving little amounts of biological material and that the experimentation time is reduced. However, this reviewer has some issues that authors need to address:
- Previous works, as stated in the introduction by the authors, have already published dual extractions from the same sample to extract metabolites and lipids (references 17-22). Although these methods have been mostly applied to untargeted approaches, authors need to highlight the novel relevant contributions of their method with respect to the already published protocols.
- In line with the previous observation, authors should comment on the advantages of the molecular composition of the solvent extraction. What is the purpose of using formic acid in the extraction mixture?
- Compared to classical lipid and metabolite extractions, it is clear that the dual extraction proposed reduces the time of sample preparation. However, to perform the targeted analysis of lipids and metabolites, the authors use four different analytical instrumentation to analyze the same sample. To this reviewer, this fact complicates the experimental part of the metabolomics workflow. I understand that metabolites and lipids need different analytical conditions, e.g. different chromatographic columns using the same LC-MS equipment, but this reviewer considers that four different equipment is too much for the same sample. Have the authors tried to use just two different LC-MS conditions to detect and quantify the same metabolites and lipids in the same sample extraction? This should be explained in the main text, so the reader understands the applicability of the protocol using the classical metabolite/lipid analytical methods.
- The authors should write about the analytical methods chosen to analyze lipids and metabolites. Why did the authors choose a HILIC column for lipid analysis and reverse-phase for amino acids?
- Please explain in the main text why HepaRG cells need to be differentiated to perform the assays.
Round 2
Reviewer 1 Report
Although several claims are toned down by the authors, it should be demonstrated advantage of the double extraction with similar protocols and not only classical extraction protocol.
As suggested by Authors Rampler et al. 2019 provides similar method for lipid and other metabolites. Also, SIMPLEX protocol published earlier do detect lipids and metabolite from central carbon metabolism. (doi:10.1074/mcp.M115.053702) Author should compare and demonstrate improvement over these previously reported double extraction protocols.
Author has used 5 methods to analyze metabolites and lipids. Splitting one sample extract in 5 parts can actually increase the amount of sample required for the analysis which is disadvantage over classical and previously reported methods.
Also, the statistical data analysis part is not clear and method for metabolite quantification need more details. Author can use scatter plot or PCA plots to show the effect of fenofibrate on the cells.
Reviewer 2 Report
The main concerns of this reviewer regarding the comparison to the already existing methods and the description of the analytical procedures have been correctly addressed by the authors and amended in the main text.
Author Response
We thank the reviewer for acknowledging the clarifications and improvements made in the manuscript regarding his previous concerns and issues.